# High baseline body mass index predicts recovery of CD4+ T lymphocytes for HIV/AIDS patients receiving long-term antiviral therapy

**Jiawen Zhu**[1,2☯], **Haimei Huang**[1☯], **Min Wang**[1], **Yun Zhang**[1], **Jinli Mo**[1,2], **Weiyi Tian**[1],
**Sumin Tan**[1], **Li Jiang**[1], **Zhihao Meng**[3]*, **Shanfang Qin**[3]*, **Chuanyi Ning**[1,2]*

**1** Nursing College, Guangxi Medical University, Nanning, Guangxi, China, **2** The Second Affiliated Hospital, Guangxi Medical University, Nanning, Guangxi, China, **3** Chest Hospital of Guangxi Zhuang Autonomous Region, Liuzhou, Guangxi, China

☯ These authors contributed equally to this work.

\* ningchuanyi@126.com (CN); qsf2271976481@163.com (SQ); chimzh@sina.com (ZM)

**Data Availability Statement:** All data generated or analyzed during this study are included in this published article. There are no publicly available data or materials.

## Abstract

The relationship between baseline BMI and CD4+ T cells during follow-up in HIV patients in China requires further evaluation. We conducted a retrospective cohort study based on adult AIDS patients who underwent or received antiretroviral therapy from 2003 to 2019 in Guangxi, China. BMI was divided into categories and compared, and after adjusting for BMI being related to the change in CD4 lymphocyte count, with normal weight as the reference group, the BMI before treatment was positively correlated with the changes in CD4+ T cells at different time periods. Among them, obese patients had significant CD4+ cell gain. In patients with pretreatment CD4+ T lymphocyte counts <200 cells/μL, a higher BMI was associated with an increased likelihood of achieving immunologic reconstitution [≥350 cells/μL: AHR: 1.02(1.01, 1.04), P = 0.004; ≥500 cells/μL: AHR: 1.03 (1.01, 1.05), P = 0.004]. Underweight in HIV patients was a risk factor for poor viral suppression [AHR: 1.24 (1.04, 1.48), P = 0.016]. Our study demonstrated that HIV/AIDS patients receiving ART with higher baseline BMI had better immune reconstitution and that baseline BMI could be an important predictor of immune reconstitution in patients receiving ART. Baseline BMI was not associated with virological failure, but a lower baseline BMI indicated poor viral suppression during follow-up.

## Introduction

In the antiretroviral therapy (ART) era, higher opportunistic infection and mortality rates are associated with immunodeficiency [1–3]. With the success of ART, the pathogen-specific immune recovery of the human body is promoted, thereby reducing the morbidity and mortality of AIDS- and non-AIDS-associated diseases [4]. CD4+ T lymphocyte counts are repeated at least once a year to monitor the efficacy after treatment, and continuous monitoring of peripheral CD4+ T cells in patients receiving long-term ART is important for understanding the disease progression of HIV infections [5, 6].

**Funding:** This study received support from the National Natural Science Foundation of China (Grant Nos. 81803295 and 81760602), the Natural Science Foundation of Guangxi (2018GXNSFAA138031), the "Thirteenth Five-Year" National Major Science and Technology Projects (2018ZX10715008–002 and 2018ZX10302104–001), the Innovation Project of Guangxi Graduate Education (YCSW2021143), and the Opening topic fund of Guangxi Key Laboratory of AIDS Prevention and Treatment (No.gklapt 201902). There was no additional external funding received for this study. The funders had no role in study design, data collection and analysis, decision to publish, or preparation of the manuscript.

**Competing interests:** The authors have declared that no competing interests exist.

**Abbreviations:** BMI, Body mass index; HIV, human immunodeficiency virus; AIDS, Acquired Immune Deficiency Syndrome; WHO, World Health Organization; ART, antiretroviral therapy; VL, viral load; HR, hazard ratio; AHR, adjusted hazard ratio; CI, confidence interval; LOESS, local polynomial regression fitting.

While the virus is completely suppressed, HIV-infected patients' CD4+ T lymphocyte count will gradually increase by 50–150 cells/mm$^3$ per year on average. The CD4+ T-cell count usually increases rapidly in the first 3 months after treatment and gradually increases over time until it reaches the normal value (>500 cells/mm$^3$) and then plateaus [7]. Younger age and lower viral load were associated with a large gain in CD4+ T cells during ART follow-up [8–10]. A previous study used a semimechanistic population model to describe the trajectories of CD4+ T cells after treatment in HIV-infected patients, suggesting that older age is a risk factor for immune reconstitution in HIV patients [11]. The model results showed that the percentage difference of patients aged ≥ 50 years achieving adequate immune reconstitution (CD4 + T-cell count > 500 cell/μL) was 15%, 21%, and 26% in the first year, fourth year, and steady-state, respectively, compared with patients aged 18–35 years. A 10-year cohort study [10] of HIV-infected injection drug users from the Asia-Pacific region reported that patients with a viral load of ≤400 copies/mL during follow-up had a greater increase in CD4 cell count over time. Decreased ten-year mean levels of CD4+ T-cell change in patients with viral loads of 401 to 100,000 copies/mL (-65.3, 95% CI: -106.6, -23.9) and >100,000 copies/mL (-121.4, 95% CI: -176.0, -66.7) were compared with patients with viral loads of ≤400 copies/mL.

Body mass index (BMI) is associated with immune reconstitution in antiviral therapy patients. Our results demonstrate that overweight HIV-infected individuals who were obese at diagnosis had greater increases in CD4 counts over time than overweight and normal weight at diagnosis [12]. A retrospective cohort study of HIV referral centres in Singapore found no association between BMI and the magnitude of change in immune recovery [13]. Several studies have reported significantly lower CD4+ cell gain after ART in obese patients than patients with normal body mass index, suggesting that obesity may adversely affect immune reconstitution [14–16]. In addition, a higher baseline BMI predicts greater gains in CD4+ T lymphocyte count in men at weeks 96 and 144 after starting ART [17]. Current research indicates that obesity is positively correlated with levels of the adipokine, leptin [18]. Leptin has been shown to directly modulate innate and adaptive immune responses [19]. Leptin plays a role in T-cell thymus development, naive CD4+ T-cell proliferation, and Ki67 expression in CD4+ T cells [20–22], and HIV patients with higher leptin levels have better immune reconstitution [23], suggesting that obesity may play a protective role in immune reconstitution in HIV patients. Elucidating the relationship between BMI and CD4+ T-cell recovery is critical for a greater understanding of the biological determinants of immune reconstitution. However, conclusions about the effect of different BMI values on immune reconstitution after antiviral therapy remain inconsistent. At present, there are relatively limited studies on the clinical characteristics and follow-up efficacy of antiviral therapy in HIV/AIDS populations with different BMI categories in China. In this study, we assessed the relationship between immune reconstitution and baseline BMI through routine follow-up medical care based on data from patients who initiated or received antiviral therapy in Guangxi from 2003 to 2019.

## Materials and methods

### Study design and study population

This was a retrospective cohort study of HIV/AIDS patients who initiated or received antiretroviral therapy in Guangxi, China, from 2003 to 2019. The study cohort included patients who received ART between January 1, 2003, and December 31, 2019, we had access to information that could identify individual participants during or after data collection. The inclusion criteria of patients were as follows: (a) age ≥ 18 years; (b) no pregnancy during baseline and follow-up; (c) baseline data included both weight and height indicators; (d) baseline included

CD4 indicators; and (e) follow-up data had at least one CD4 record. **Fig 1** shows a flowchart of the patient inclusion criteria.

## Data collection and definition

Data were obtained from the China AIDS Comprehensive Response Information Management System (CRIMS), including all participants' general demographic characteristics and clinical laboratory test indicators. In our study, the grades of BMI were classified according to the standards established by the health industry standards of the People's Republic of China: underweight (BMI<18.5 kg/m$^2$), normal (18.5 kg/m$^2\leq$BMI<24 kg/m$^2$), overweight (24 kg/m$^2\leq$BMI<28 kg/m$^2$), and obese (BMI$\geq$28 kg/m$^2$). BMI was calculated as weight (kg)/height (m)$^2$. Opportunistic infections included skin lesions, oral thrush, oral leukoplakia, persistent diarrhoea, persistent or intermittent fever, recurrent severe bacterial infections, disseminated nontuberculous infections, oesophageal candidiasis, and extrapulmonary cryptococcal infections. Changes in CD4 lymphocyte counts during follow-up were calculated as the difference between the baseline CD4 and CD4 lymphocyte counts at different time points (within ± 3

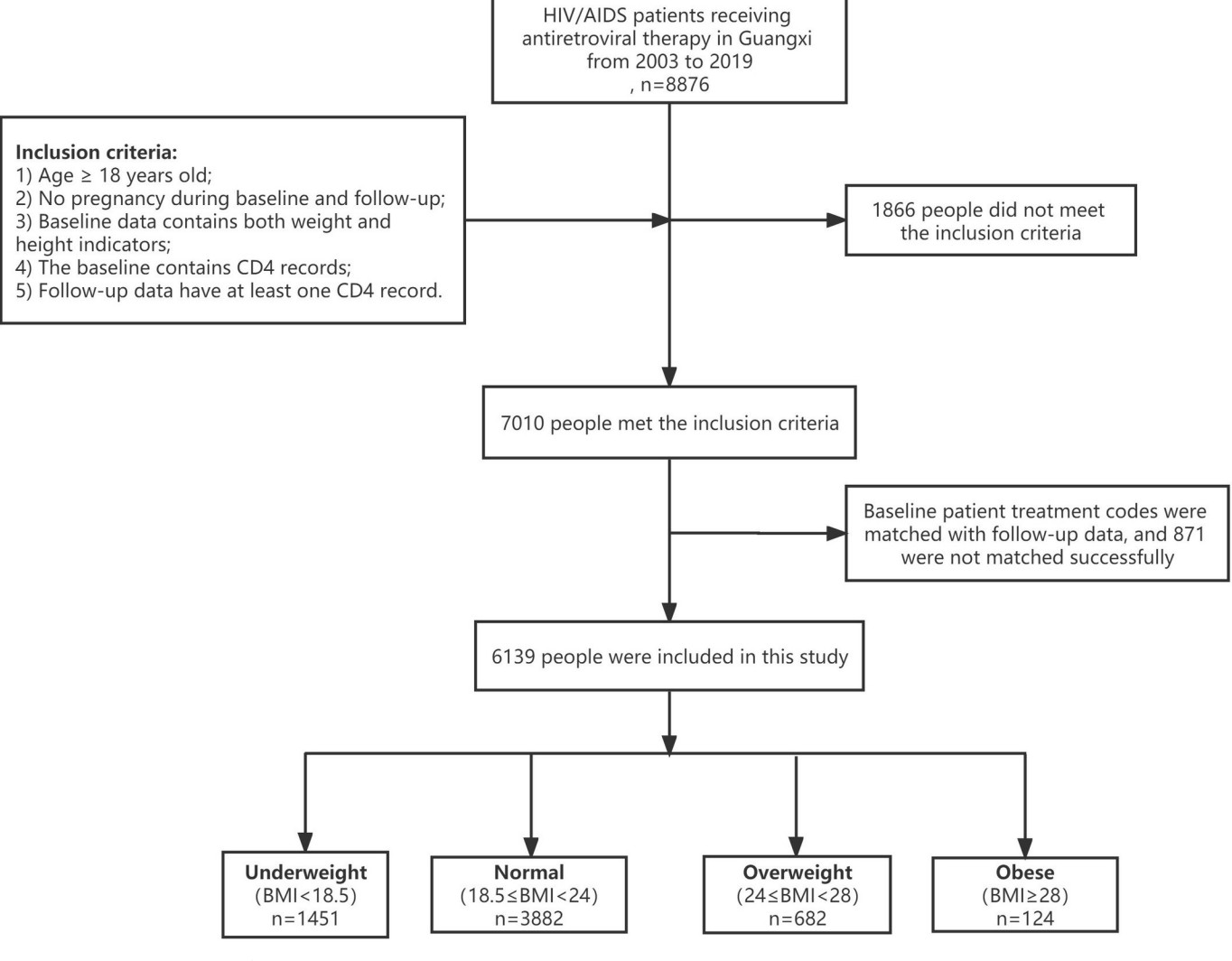

**Fig 1. Flow chart of the patients' inclusion criteria.**

months of each time point). Virological failure was defined as two consecutive plasma HIV RNA >400 copies/ml, and poor viral suppression was defined as a viral load >400 copies/mL during subsequent follow-up in patients with >6 months of follow-up.

## Statistical analysis

Differences in demographic and clinical characteristics of patients stratified by BMI were compared using the Pearson $\chi^2$ test (discrete variable) and the Kruskal–Wallis test (continuous variable). Univariate and multivariate linear regression analyses were performed to analyse the association between baseline variables and changes in CD4 lymphocyte counts in the first, third, fifth, and eighth years, and the normal body mass index group was used as the reference object. The discrete variables at baseline were treated as dummy variables. Assuming a potential impact of viral suppression on the recovery of CD4 lymphocyte counts during follow-up, virological failure was included in the linear regression model.

To assess the effect of BMI on increasing CD4+ T-cell counts and reaching thresholds of 350 cells/μL and 500 cells/μL, we used Kaplan–Meier plots to compare the rates of CD4 lymphocyte count recovery for reaching ≥ 350 cells/μL and ≥ 500 cells/μL in patients with different BMI categories.

The analysis cohort was limited to patients with a baseline CD4+ lymphocyte count <200 cells/μL, and the Cox proportional risk model was used to determine the independent influence of the baseline CD4 lymphocyte count reaching the threshold for the first time (BMI as a continuous variable). Variables with statistical significance (P<0.05) or clinical significance of the univariate analysis were included in the multivariate equation (LR) for analysis. In addition, Cox proportional hazards models were used to assess the impact of the risk of first reaching the event endpoint (CD4 ≥ 500 cells/μL) in each BMI category. At the same time, an unadjusted analysis (crude model), minimally adjusted analysis (Model I adjusted for sex and age), and fully adjusted analysis (Model II adjusted for sex, age, marital status, route of infection, current WHO stage, baseline CD4 count, VL (log10 copies/ml), presence of pulmonary tuberculosis in the last year, presence of opportunistic infections, initial treatment regimen, length of delay from diagnosis to treatment, and previous use of cotrimoxazole) were used to enhance the reliability of the results according to the Statement for Reporting Observational Studies in Epidemiology (STROBE). To avoid bias caused by differences between BMI levels, sensitivity analysis was used to further verify the reliability of the results, and trend tests were performed by BMI categories.

Finally, we used local polynomial regression fitting (LOESS) to assess the relationship between baseline BMI and viral load and assessed the association of baseline BMI with virological failure and poor viral suppression using a Cox proportional hazards model.

All analyses were performed using R (version 4.0.1; www.r-project.org) software and SPSS version 26.0 (SPSS Inc., Chicago, USA). P<0.05 was considered statistically significant.

## Ethics statement

This study was reviewed and approved by the Human Research Ethics Committee of Chest Hospital of Guangxi Zhuang Autonomous Region (Ethical review No. 2018–006), and individual consent was waived by the ethics committee.

## Results

### Baseline characteristics of different BMI groups

Statistical analysis was performed on 6139 patients (28167.4 person-years). Among these patients, BMI values were calculated using records of their height and weight and were classified

according to the standards established by the health industry standards of the People's Republic of China. The demographic and clinical characteristics are shown in **Table 1**. A total of 1451 (23.6%) patients were underweight, 3882 (63.2%) were normal weight, 682 (11.1%) were overweight, and 124 (2%) were obese. The majority of patients receiving ART were 4183 males (68.1%), and the females were classified more often as underweight (35.5%) than other BMI groups (p = 0.005); the median age of the patients was 40 (IQR: 31–52); 3766 (61.3%) were married or cohabiting, and the main route of infection was sexual transmission (93.1%); most of the underweight patients were in WHO clinical stage IV (41.6%) and had pulmonary tuberculosis

**Table 1. Demographic characteristics of HIV patients receiving ART by BMI groups.**

| Cohort at ART initiation | Underweight (BMI<18.5) n = 1451 | BMI Normal (18.5≤BMI<24) n = 3882 | Overweight (24≤BMI<28) n = 682 | Obese (BMI≥28) n = 124 | P-value | Overall |
|---|---|---|---|---|---|---|
| **Gender** | | | | | 0.005 | |
| Male (%) | 936(64.5%) | 2702(69.6%) | 462(67.7%) | 83(66.9%) | | 4183(68.1%) |
| Female (%) | 515(35.5%) | 1180(30.4%) | 220(32.3%) | 41(33.1%) | | 1956(31.9%) |
| **Age (year) Median, IQR** | 38(29–53) | 40(31–52) | 42(34–53) | 39(30–49) | <0.001 | 40(31–52) |
| **Marital status** | | | | | <0.001 | |
| unmarried | 427(29.4%) | 939(24.2%) | 121(17.7%) | 34(27.4%) | | 1521(24.8%) |
| married or living together | 812(56.0%) | 2412(62.1%) | 469(68.8%) | 73(58.9%) | | 3766(61.3%) |
| divorced or separated | 79(5.4%) | 286(7.4%) | 51(7.5%) | 10(8.1%) | | 426(6.9%) |
| Widowed and others | 133(9.2%) | 245(6.3%) | 41(6.0%) | 7(5.6%) | | 426(6.9%) |
| **Transmission route** | | | | | <0.001 | |
| blood transmission | 4(0.3%) | 14(0.4%) | 1(0.1%) | 0(0.0%) | | 19(0.3%) |
| Intravenous drug use | 85(5.9%) | 158(4.1%) | 14(2.1%) | 3(2.4%) | | 260(4.2%) |
| sexually transmitted | 1320(91.0%) | 3635(93.6%) | 642(94.1%) | 118(95.2%) | | 5715(93.1%) |
| other | 42(2.9%) | 75(1.9%) | 25(3.7%) | 3(2.4%) | | 145(2.4%) |
| **WHO clinical stage** | | | | | <0.001 | |
| I | 436(30.0%) | 1987(51.2%) | 471(69.1%) | 86(69.4%) | | 2980(48.5%) |
| II | 155(10.7%) | 412(10.6%) | 72(10.6%) | 11(8.9%) | | 650(10.6%) |
| III | 257(17.7%) | 533(13.7%) | 53(13.7%) | 13(10.5%) | | 856(13.9%) |
| IV | 603(41.6%) | 950(24.5%) | 86(12.6%) | 14(11.3%) | | 1653(26.9%) |
| **CD4+ T-cell count (cells/ml), Median, IQR** | 57(16–206) | 176(47–299) | 255(141–347) | 298(164–385 | <0.001 | 162(37–293) |
| **VL(log10 copies/ml), Median, IQR** | 5.11(4.62–5.58) | 4.88(4.34–5.38) | 4.76(4.22–5.22) | 4.47(3.97–5.05) | <0.001 | 4.92(4.37–5.41) |
| **Mtb infection in the recent year** | | | | | <0.001 | |
| Yes | 315(21.7%) | 456(11.7%) | 33(4.8%) | 7(5.6%) | | 811(13.2%) |
| No | 1131(77.9%) | 3398(87.5%) | 646(94.7%) | 116(93.5%) | | 5291(86.2%) |
| Unknown | 5(0.3%) | 28(0.7%) | 3(0.4%) | 1(0.8%) | | 37(0.6%) |
| **Opportunistic infection** | | | | | <0.001 | |
| Yes | 843(58.1%) | 1405(36.2%) | 126(18.5%) | 19(15.3%) | | 2393(39.0%) |
| No | 608(41.9%) | 2477(63.8%) | 556(81.5%) | 105(84.7%) | | 3746(61.0%) |
| **Initial treatment plan** | | | | | 0.184 | |
| PI-based | 105(7.2%) | 329(8.5%) | 72(10.6%) | 16(12.9%) | | 522(8.5%) |
| NNRTI-based | 1324(91.2%) | 3503(90.2%) | 603(88.4%) | 108(87.1%) | | 5538(90.2%) |
| NRTI-only | 11(0.8%) | 31(0.8%) | 5(0.7%) | 0 | | 47(0.8%) |
| Other | 11(0.8%) | 19(0.5%) | 2(0.3%) | 0 | | 32(0.5%) |
| **Time from diagnosis to treatment delay (≤3 months)** | 1134(78.2%) | 2920(75.2%) | 497(72.9%) | 94(75.8%) | 0.042 | 4645(75.7%) |
| **Cotrimoxazole use before baseline (yes)** | 787(54.2%) | 1397(36.0%) | 154(22.6%) | 23(18.5%) | <0.001 | 2361(38.5%) |

in the last year before antiviral therapy (21.7%). The incidence of opportunistic infections in the underweight group was higher (58.1%) than in other BMI groups (P<0.001); the initial antiretroviral regimen of all patients was mainly based on NNRTI (90.2%), and the duration of diagnosis to treatment delay ≤3 months ratio was as high as 75.7%; for the previous use of cotrimoxazole, use in the underweight group was higher than in the other groups (54.2%). Laboratory examination results showed that the median CD4 lymphocyte count in the overweight/ obese group was higher than that in the underweight and normal groups (P<0.05); the higher the BMI category was, the lower the median VL (log10 copies/ml) count (P<0.05).

## Association of baseline BMI groupings with changes in CD4 lymphocyte counts during follow-up

HIV/AIDS patients receiving antiretroviral therapy were followed for an average of 4.6 years, with a median follow-up of 4.6 (IQR: 1.5–7.4) years and a maximum follow-up of 12 years. Most patients were followed for ≤8 years. **Fig 2** shows immune recovery by BMI categories from the start of ART to the 8th year of follow-up, with the greatest increase in CD4 lymphocyte counts in the first year after treatment. The higher the BMI category was, the higher the CD4 lymphocyte count, which gradually increased with the duration of follow-up. CD4 lymphocyte counts in the first year (4777 individuals), third year (3422 individuals), fifth year (2582 individuals), and eighth year (874 individuals) were used to calculate the change in concentration. The influencing factors for CD4 level changes in different years were assessed by multivariate linear regression. Virologic failure was assumed to potentially elevate CD4 cell count, so virologic failure was included as a covariate in the model. Virologic treatment failure was defined as two consecutive plasma results of HIV RNA > 400 copies/ml after 6 months of antiviral therapy. A total of 300 patients failed. The results of **Table 2** showed that after adjusting for BMI, which was related to the change in CD4 lymphocyte count, with normal weight as the reference group, the gain in CD4+ T cells in overweight patients in the first year was not significant [8.70(-6.17, 23.56), P = 0.250], and the patients with higher BMI in the subsequent period had higher changes in CD4 lymphocyte count (P<0.05), all of which were statistically significant. Obese patients had significant CD4+ cell gain [1st year: 72.27 (38.96, 105.58), P<0.001; 3rd year: 78.25 (28.40, 128.09), P = 0.002; 5th year: 113.37 (48.37, 178.37), P<0.001; 8th year:187.24(60.12, 314.36), P = 0.004]. In addition, age, the occurrence of virologic failure, and sex were related to the change in CD4+ T cells. Taking the fifth year as an example, older age [-3.36 (-4.06, -2.65), P<0.001] and the occurrence of virologic failure [-115.81 (-150.32, -81.29), P<0.001] were negative factors for CD4+ T-cell gain. CD4+ T-cell concentration was significantly increased in females compared with males [48.28 (30.15, 66.41), P<0.001]. We added the analysis to compare the CD4 changes between the HIV positive, AIDS with AIDS defining symptoms and AIDS without AIDS defining symptoms. The data analysis results showed that there was no correlation between each group for CD4 changes, and the difference was not statistically significant. And we selected patients defined as AIDS, and divided them into two groups according to whether they have AIDS defining symptoms or not. We used a linear regression model to analyze and compare the changes of CD4 in patients without AIDS defining symptoms versus patients with AIDS defining symptoms. Results also shown that there was no statistical difference between these two groups. Please see more details in the **S1 Table**.

## CD4 lymphocyte count <200 cells/μL recovery rate after starting antiviral therapy and its influencing factors

At baseline, 3474 patients had baseline CD4+ T-cell counts <200 cells/μL, 1994 patients (57.4%) had CD4 lymphocyte counts that first recovered to ≥350 cells/μL during follow-up,

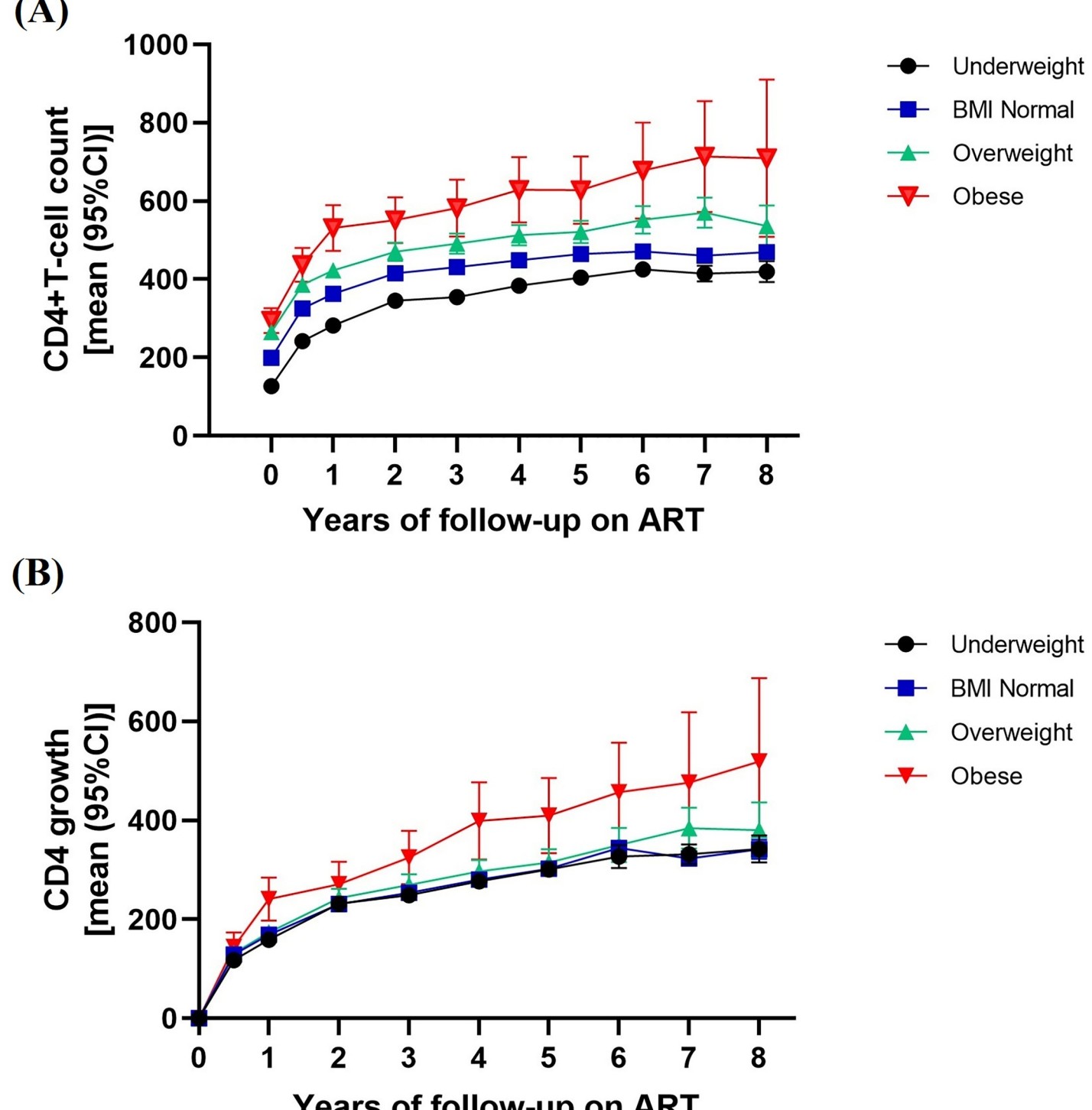

**Fig 2. Immune recovery and growth in different BMI categories during 8-year ART treatment.** (A) shows immune recovery by BMI categories from the start of ART to the 8th year of follow-up; (B) shows Mean increase in CD4 cell count from the start of ART to the 8th year of follow-up.

and 1185 patients (34.1%) recovered to ≥500 cells/μL. As a result, the total rate of CD4 + count recovery at the thresholds (≥350 cells/μL, ≥500 cells/μL) was 14.7/100 person-

**Table 2. Univariate and multivariate analysis of CD4 changes at different time points of antiretroviral therapy.**

| Variable | Changes in CD4 cell counts in the 1st year of antiviral therapy | | | | Changes in CD4 cell counts in the 3rd year of antiviral therapy | | | | Changes in CD4 cell counts in the 5th year of antiviral therapy | | | | Changes in CD4 cell counts in the 8th year of antiviral therapy | | | |
| --- | --- | --- | --- | --- | --- | --- | --- | --- | --- | --- | --- | --- | --- | --- | --- | --- |
| | Univariate analysis | | Multivariable analysis | | Univariate analysis | | Multivariable analysis | | Univariate analysis | | Multivariable analysis | | Univariate analysis | | Multivariable analysis | |
| | 95%CI | P | 95%CI | P | 95%CI | P | 95%CI | P | 95%CI | P | 95%CI | P | 95%CI | P | 95%CI | P |
| **BMI** | | | | | | | | | | | | | | | | |
| BMI<18.5 | -10.32 (-21.45, 0.80) | 0.07 | -16.74 (-28.14, -5.33) | 0.004 | -4.34 (-19.34, 10.65) | 0.57 | -21.80 (-36.83, -6.78) | 0.004 | -1.25 (-20.87, 18.38) | 0.901 | -29.30 (-48.63, -9.97) | 0.003 | 0.35 (-34.63, 35.33) | 0.984 | -44.54 (-77.79, -11.3) | 0.009 |
| (18.5≤BMI<24) (reference) | 1 | | 1 | | 1 | | 1 | | 1 | | 1 | | 1 | | 1 | |
| (24≤BMI<28) | 4.85 (-10.05, 19.74) | 0.52 | 8.70(-6.17, 23.56) | 0.250 | 16.45 (-4.59, 37.48) | 0.125 | 27.80 (7.38, 48.21) | 0.008 | 12.70 (-16.03, 41.43) | 0.386 | 32.75 (5.16, 60.34) | 0.020 | 38.2 (-19.05, 95.45) | 0.191 | 70.66 (17.78, 123.55) | 0.009 |
| (BMI≥28) | 72.16 (38.50, 105.82) | <0.001 | 72.27 (38.96, 105.58) | <0.001 | 71.81 (20.12, 123.51) | 0.006 | 78.25 (28.40, 128.09) | 0.002 | 107.81 (39.59, 176.07) | 0.002 | 113.37 (48.37, 178.37) | <0.001 | 177.35 (39.11, 315.59) | 0.012 | 187.24 (60.12, 314.36) | 0.004 |
| **Gender (ref:Male)** | 17.76 (7.96, 27.6) | <0.001 | 18.34 (7.86, 28.83) | <0.001 | 33.60 (20.46, 46.74) | <0.001 | 34.48 (20.65, 48.31) | <0.001 | 38.82 (21.43, 56.21) | <0.001 | 48.28 (30.15, 66.41) | <0.001 | 29.82 (2.37, 62.01) | 0.019 | 53.8 (20.81, 86.79) | 0.001 |
| **Age** | -1.42 (-1.76, -1.09) | <0.001 | -1.44 (-1.83, -1.04) | <0.001 | -2.62 (-3.09, -2.15) | <0.001 | -8.78 (-3.15, -2.09) | <0.001 | -3.39 (-4.02, -2.75) | <0.001 | -3.36 (-4.06, -2.65) | <0.001 | -3.32 (-4.55, -2.1) | < 0.001 | -3.56 (-4.86, -2.26) | < 0.001 |
| **Marital status** | | | | | | | | | | | | | | | | |
| unmarried (reference) | 1 | | 1 | | 1 | | 1 | | 1 | | 1 | | 1 | | 1 | |
| married or living together | -22.46 (-33.65, -11.28) | <0.001 | -14.22 (-26.93, -1.52) | 0.028 | -22.99 (-39.13, -6.85) | 0.005 | -10.44 (-27.76, 6.88) | 0.237 | -38.48 (-61.13, -15.83) | <0.001 | -22.81 (-46.36, 0.73) | 0.06 | -13.73 (-55.47, 28.02) | 0.519 | 10.5 (-31.31, 52.31) | 0.622 |
| divorced or separated | -34.52 (-54.95, -14.10) | <0.001 | -24.32 (-45.57, -3.07) | 0.025 | -25.47 (-55.01, 4.06) | 0.091 | -6.66 (-36.35, 23.03) | 0.66 | -37.85 (-78.77, 3.08) | 0.070 | -18.38 (-58.74, 21.98) | 0.372 | -28.8 (-111.76, 54.16) | 0.496 | 23.09 (-54.85, 101.03) | 0.561 |
| Widowed and others | -36.38 (-56.60, -16.15) | <0.001 | -13.84 (-36.59, 8.90) | 0.233 | -55.16 (-83.23, -27.08) | <0.001 | -13.15 (-43.50, 17.21) | 0.396 | -59.04 (-97.53, -20.54) | 0.003 | -10.56 (-51.23, 30.11) | 0.611 | -4.42 (-75.55, 66.71) | 0.903 | -10.47 (-21.23, 72.17) | 0.167 |
| **Transmission route** | | | | | | | | | | | | | | | | |
| blood transmission (reference) | 1 | | 1 | | 1 | | 1 | | 1 | | 1 | | 1 | | 1 | |
| Intravenous drug use | -28.86 (-109.42, 51.70) | 0.483 | -31.93 (-111.38, 47.51) | 0.431 | -67.72 (-178.31, 42.87) | 0.23 | -72.2 (-178.66, 34.25) | 0.184 | 40.87 (-99.70, 181.45) | 0.569 | -10.54 (-144.39, 123.30) | 0.877 | -46.02 (-352.48, 99.55) | 0.456 | -40.47 (-70.92, 10.03) | 0.334 |
| sexually transmitted | 1.26 (-76.12, 78.65) | 0.974 | -6.54 (-82.84, 69.76) | 0.867 | -31.38 (-137.97, 75.21) | 0.564 | -37.26 (-139.78, 65.25) | 0.476 | 81.54 (-53.71, 216.79) | 0.237 | 36.42 (-92.55, 30.11) | 0.579 | -36.36 (-636.62, 35.9) | 0.222 | -32.04 (-61.64, 26.43) | 0.411 |
| other | 34.60 (-47.97, 117.17) | 0.411 | 25.45 (-55.98, 106.88) | 0.54 | -2.66 (-116.85, 111.52) | 0.964 | -13.99 (-123.86, 95.88) | 0.803 | 74.44 (-43.15, 248.77) | 0.167 | 60.20 (-78.87, 199.27) | 0.396 | -32.5 (-65.8, 33) | 0.134 | -52.9 (-68.26, 9.74) | 0.861 |

(Continued)

**Table 2.** (Continued)

| Variable | Changes in CD4 cell counts in the 1st year of antiviral therapy | | | | Changes in CD4 cell counts in the 3rd year of antiviral therapy | | | | Changes in CD4 cell counts in the 5th year of antiviral therapy | | | | Changes in CD4 cell counts in the 8th year of antiviral therapy | | | |
| --- | --- | --- | --- | --- | --- | --- | --- | --- | --- | --- | --- | --- | --- | --- | --- | --- |
| | Univariable analysis | | Multivariable analysis | | Univariable analysis | | Multivariable analysis | | Univariable analysis | | Multivariable analysis | | Univariable analysis | | Multivariable analysis | |
| | 95%CI | P | 95%CI | P | 95%CI | P | 95%CI | P | 95%CI | P | 95%CI | P | 95%CI | P | 95%CI | P |
| **WHO clinical stage** | | | | | | | | | | | | | | | | |
| I (reference) | 1 | | 1 | | 1 | | 1 | | 1 | | 1 | | 1 | | 1 | |
| II | -3.23 (-18.59, 12.14) | 0.68 | -0.94 (-17.39, 15.51) | 0.91 | -12.80 (-33.42, 7.81) | 0.223 | -25.78 (-47.41, -4.15) | 0.019 | 9.30 (-18.41, 37.01) | 0.510 | -22.82 (-51.76, 6.12) | 0.122 | -40.9 (-78, 57) | 0.133 | -20.04 (-72.05, 31.96) | 0.45 |
| III | -1.16 (-15.22, 12.89) | 0.87 | 9.14 (-8.78, 27.07) | 0.317 | 3.29 (-15.63, 22.21) | 0.733 | -2.21 (-25.58, 21.17) | 0.853 | 13.06 (-11.62, 37.74) | 0.299 | -11.03 (-41.27, 19.21) | 0.474 | -22.4 (-44.54, 55.24) | 0.245 | 14.24 (-39.88, 68.35) | 0.606 |
| IV | -11.14 (-22.20, 0.08) | 0.05 | -6.19 (-24.08, 11.68) | 0.497 | 16.23 (1.27, 31.20) | 0.034 | -8.78 (-32.46, 14.91) | 0.468 | 28.85 (8.90, 48.79) | 0.005 | -24.41 (-55.48, 6.67) | 0.124 | -39.45 (-67.22, 45.3) | 0.487 | -1.56 (-57.35, 54.22) | 0.956 |
| **CD4+ T-cell count** | -0.01 (-0.04, 0.02) | 0.43 | -0.06 (-0.09, -0.02) | 0.001 | -0.12 (-0.16, -0.08) | <0.001 | -0.19 (-0.24, -0.13) | <0.001 | -0.24 (-0.30, -0.18) | <0.001 | -0.33 (-0.41, -0.25) | <0.001 | -0.55 (-0.68, -0.43) | <0.001 | -0.61 (-0.78, -0.44) | <0.001 |
| **VL(log10 copies/ml)** | 7.06 (2.72, 11.39) | 0.001 | 10.63 (6.13, 15.13) | <0.001 | 9.01 (3.02, 15.00) | 0.003 | 10.42 (4.30, 16.54) | <0.001 | 10.34 (2.10, 18.58) | 0.014 | 8.76 (0.43, 17.10) | 0.039 | 19.22 (-5.09, 33.36) | 0.118 | 11.25 (-3.29, 24.79) | 0.123 |
| **Cotrimoxazole use before baseline (ref: yes)** | 9.31 (-0.11, 18.73) | 0.05 | 10.93 (-1.13, 22.99) | 0.076 | -10.45 (-23.15, 2.26) | 0.107 | 12.88 (-3.02, 28.77) | 0.112 | -29.38 (-46.22, -12.53) | <0.001 | 7.11 (-13.86, 28.08) | 0.506 | -87.8 (-118.32, -57.29) | <0.001 | -22.74 (-60.06, 14.58) | 0.232 |
| **Mtb infection in the recent year** | | | | | | | | | | | | | | | | |
| Yes (reference) | 1 | | 1 | | 1 | | 1 | | 1 | | 1 | | 1 | | 1 | |
| No | 18.53 (5.07, 31.98) | 0.007 | 11.24 (-3.63, 26.11) | 0.138 | 0.79 (-16.90, 18.47) | 0.93 | 3.40 (-15.48, 22.27) | 0.724 | -5.33 (-28.14, 17.48) | 0.647 | -1.50 (-25.75, 22.74) | 0.903 | -62.18 (-100.92, -23.43) | 0.002 | -36.46 (-75.98, 3.05) | 0.07 |
| Unknown | 11.26 (-53.67, 76.18) | 0.734 | 2.44 (-61.85, 66.74) | 0.941 | 41.51 (-42.56, 125.58) | 0.333 | 25.50 (-55.66, 106.66) | 0.538 | -27.02 (-139.28, 85.24) | 0.637 | -47.74 (-155.06, 59.58) | 0.383 | 36.38 (-63.6, 58.17) | 0.678 | 37.27 (-64.74, 54.8) | 0.468 |
| **Initial treatment plan** | | | | | | | | | | | | | | | | |
| PI-based (reference) | 1 | | 1 | | 1 | | 1 | | 1 | | 1 | | 1 | | 1 | |
| NNRTI-based | -18.62 (-35.77, -1.48) | 0.033 | -19.99 (-37.13, -2.86) | 0.022 | -4.65 (-29.44, 20.14) | 0.713 | -14.42 (-38.73, 9.89) | 0.245 | 39.69 (1.56, 77.82) | 0.041 | 8.63 (-28.36, 45.63) | 0.647 | 42.49 (-35.03, 120.01) | 0.282 | -26.68 (-101.81, 48.46) | 0.486 |
| NRTI-only | -51.28 (-107.59, 5.03) | 0.074 | -43.33 (-99.01, 12.34) | 0.127 | -50.90 (-138.82, 37.02) | 0.256 | -47.65 (-132.35, 37.05) | 0.27 | -64.80 (-230, 100.71) | 0.442 | -78.27 (-235.93, 79.40) | 0.330 | 95.47 (-235.4, 426.34) | 0.571 | -74.34 (-237.68, 386.37) | 0.64 |
| Other | 60.73 (-70.37, 191.83) | 0.364 | 63.71 (-65.46, 192.88) | 0.334 | -222.80 (-592.38, 146.79) | 0.237 | -289.33 (-645.28, 66.62) | 0.111 | -211.37 (639.72, 216.97) | 0.333 | -332.52 (-740.70, 75.66) | 0.110 | - | - | - | - |

*(Continued)*

Table 2. (Continued)

| Variable | Changes in CD4 cell counts in the 1st year of antiviral therapy | | | | Changes in CD4 cell counts in the 3rd year of antiviral therapy | | | | Changes in CD4 cell counts in the 5th year of antiviral therapy | | | | Changes in CD4 cell counts in the 8th year of antiviral therapy | | | |
|---|---|---|---|---|---|---|---|---|---|---|---|---|---|---|---|---|
| | Univariable analysis | | Multivariable analysis | | Univariable analysis | | Multivariable analysis | | Univariable analysis | | Multivariable analysis | | Univariable analysis | | Multivariable analysis | |
| | 95%CI | P | 95%CI | P | 95%CI | P | 95%CI | P | 95%CI | P | 95%CI | P | 95%CI | P | 95%CI | P |
| Opportunistic infection (ref:No) | -7.42 (-16.86, 2.01) | 0.123 | 1.96 (-12.35, 16.27) | 0.788 | 17.87 (5.17, 30.56) | 0.006 | 21.28 (2.50, 40.07) | 0.026 | 32.99 (16.19, 49.80) | <0.001 | 21.81 (0.07, 49.55) | 0.049 | 65.02 (-33.99, 96.06) | 0.116 | 6.31 (-36.02, 48.63) | 0.77 |
| Virologic failure (ref:No) | -43.63 (-64.50, -22.76) | <0.001 | -45.44 (-66.18, -24.69) | <0.001 | -97.09 (-123.97, -70.22) | <0.001 | -101.12 (-127.33, -74.91) | <0.001 | -101.61 (-137.44, -65.77) | <0.001 | -115.81 (-150.32, -81.29) | <0.001 | -94.77 (-162.78, -26.75) | 0.006 | -141.36 (-204.63, -78.08) | < 0.001 |
| Time from diagnosis to treatment delay (ref:≤3 months) | -4.55 (-15.08, 5.99) | 0.398 | -5.57 (-16.33, 5.19) | 0.31 | 10.08 (-4.00, 24.16) | 0.16 | 15.40 (1.21, 29.59) | 0.033 | 4.41 (-14.14, 22.97) | 0.641 | 18.41 (-0.10, 36.92) | 0.051 | -7.74 (-43.03, 27.55) | 0.667 | 36.87 (-2.04, 71.69) | 0.138 |

years and 7.8/100 person-years, respectively. Kaplan–Meier analysis was used to calculate the median time and cumulative risk rate of CD4+ T-cell recovery to the threshold in patients with different BMI categories, and the log-rank test was used for further comparison between groups. **Fig 3** shows that the median time to recovery of CD4+ T-cell counts to ≥350 cells/μL in different BMI categories (Underweight, Normal, Overweight, Obese) was 5.97, 5.64, 5.38 and 4.42 years, respectively. The median time to recovery to ≥500 cells/μL was 7.18, 7.03, 6.28 and 5.50 years, respectively. There was a significant difference in the cumulative risk of CD4+ T-cell recovery to the threshold between different BMI categories (log-rank test, P<0.001; **Fig 3**). Multiple comparisons were performed on the cumulative risk rate of CD4+ T cells returning to the threshold in different BMI categories, in which the cumulative risk rate of CD4+ T cells returning to the threshold ($\geq$ 350 cells/μL, $\geq$ 500 cells/μL) was significantly higher in the obese group than in the other groups (P < 0.05), and there was no significant difference in the cumulative risk rate of CD4+ T cells returning to the threshold ($\geq$ 500 cells/μL) between the underweight and normal BMI groups (P = 0.172).

Univariate and multivariate analyses identified the relevant variables for patients that return to the threshold (**Table 3**). The results showed that BMI, sex, age, WHO clinical stage, baseline CD4 lymphocyte count, initial treatment plan, and the occurrence of

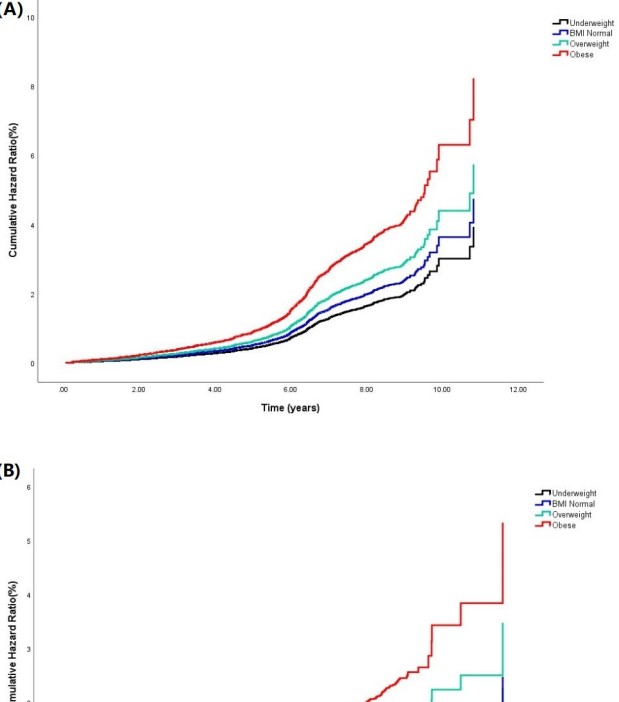

**Fig 3. Kaplan-Meier survival curves of patients with different BMI categories (underweight, BMI normal, overweight, obese) with baseline CD4+ T cells <200 cells/μL recovering to threshold.** (A) Cumulative risk rate of recovery of CD4+ T cells to ≥350 cells/μL in HIV-infected patients; (B) Cumulative risk rate of recovery of CD4+ T cells to ≥500 cells/μL in HIV-infected patients.

**Table 3. The associated factors of CD4+ T cell count recovery to the endpoints (≥ 350, 500 cells/μL).**

| | CD4+T cell count recovery to ≥ 350cells/μL | | CD4+T cell count recovery to ≥ 500cells/μL | |
|---|---|---|---|---|
| | HR(95%CI) | AHR(95%CI) | HR(95%CI) | AHR(95%CI) |
| **BMI** | 1.04(1.02, 1.05) | 1.02(1.01, 1.04) | 1.04(1.02, 1.06) | 1.03(1.01, 1.05) |
| **Gender (ref:Male)** | 1.36(1.24, 1.49) | 1.21(1.09, 1.34) | 1.60(1.43, 1.80) | 1.38(1.22, 1.56) |
| **Age** | 0.99(0.99, 1.00) | 0.99(0.99, 1.00) | 0.98(0.98, 0.99) | 0.98(0.97, 0.98) |
| **Marital status** | | | | |
| unmarried (reference) | 1 | 1 | 1 | 1 |
| married or living together | 0.80(0.72, 0.90) | 0.79(0.70, 0.90) | 0.89(0.77, 1.03) | - |
| divorced or separated | 1.13(1.10, 1.66) | 1.10(0.90, 1.36) | 1.04(0.79, 1.36) | - |
| Widowed and others | 0.86(0.70, 1.05) | 0.91(0.72, 1.14) | 0.93(0.72, 1.21) | - |
| **Transmission route** | | | | |
| blood transmission (reference) | 1 | 1 | 1 | 1 |
| Intravenous drug use | 0.33(0.16, 0.68) | - | 0.33(0.12, 0.91) | - |
| sexually transmitted | 0.34(0.17, 0.69) | - | 0.46(0.17, 1.23) | - |
| other | 0.30(0.14, 0.64) | - | 0.58(0.21, 1.65) | - |
| **WHO clinical stage** | | | | |
| I (reference) | 1 | 1 | 1 | 1 |
| II | 0.64(0.55, 0.74) | 0.75(0.64, 0.87) | 0.68(0.56, 0.82) | 0.79(0.65, 0.97) |
| III | 0.68(0.59, 0.77) | 0.93(0.81, 1.07) | 0.67(0.56, 0.72) | 1.00(0.83, 1.21) |
| IV | 0.61(0.55, 0.69) | 0.97(0.85, 1.11) | 0.64(0.56, 0.74) | 1.10(0.93, 1.31) |
| **CD4+ T-cell count** | 1.01(1.00, 1.01) | 1.00(1.00, 1.01) | 1.01(1.00, 1.01) | 1.01(1.00, 1.01) |
| **VL (log10 copies/ml)** | 1.01(0.96, 1.06) | - | 0.97(0.92, 1.03) | - |
| **Cotrimoxazole use before baseline (ref:yes)** | 1.28(1.16, 1.40) | - | 1.23(1.09, 1.38) | - |
| **Mtb infection in the recent year** | | | | |
| Yes (reference) | 1 | 1 | 1 | 1 |
| No | 1.17(1.05, 1.30) | - | 1.21(1.05, 1.40) | - |
| Unknown | 1.65(1.02, 2.68) | - | 1.38(0.73, 2.60) | - |
| **Initial treatment plan** | | | | |
| PI-based (reference) | 1 | 1 | 1 | 1 |
| NNRTI-based | 0.57(0.47, 0.69) | 0.59(0.49, 0.72) | 0.60(0.46, 0.77) | 0.62(0.48, 0.80) |
| NRTI-only | 0.82(0.49, 1.37) | 1.02(0.61, 1.71) | 0.46(0.20, 1.073) | 0.59(0.25, 1.36) |
| Other | 0.92(0.29, 2.91) | 0.85(0.27, 2.7) | 0.71(0.10, 5.01) | 0.63(0.09, 4.58) |
| **Opportunistic infection (ref:No)** | 0.69(0.63, 0.76) | - | 0.70(0.62, 0.79) | - |
| **Virologic failure (ref:No)** | 0.76(0.62, 0.91) | 0.71(0.58, 0.86) | 0.68(0.52, 0.88) | 0.65(0.50, 0.84) |
| **Time from diagnosis to treatment delay (ref:≤3 months)** | 1.16(1.04, 1.29) | - | 1.28(1.12, 1.46) | - |

virological failure were still independent factors for the recovery of CD4 lymphocyte count to the threshold in patients. Among them, the adjusted model showed that higher BMI was associated with an increased likelihood of achieving immunologic reconstitution [≥350 cells/μL: AHR: 1.02(1.01, 1.04), P = 0.004; ≥500 cells/μL: AHR: 1.03 (1.01, 1.05), P = 0.004]. We summarise the association between the recovery of CD4+ T cells to a threshold (≥500 cells/μL) and BMI in patients (**S2 Table**). The core results of the three models were consistent, and CD4 lymphocyte count recovery was positively correlated with BMI. For the purpose of sensitivity analysis, we also handled BMI as a categorical variable, and changes in the corresponding effect sizes (1, 1.03, 1.45, 1.68) of BMI categories indicated that with increasing BMI, the risk trend of CD4 lymphocyte count returning to ≥ 500 cells/μL gradually increased (P for trend < 0.001).

### Association of baseline BMI with viral load

A Cox proportional hazard model was used to estimate the relationship between baseline BMI and virological treatment failure, and univariate analysis showed that the baseline BMI value was not associated with virological failure. Multivariate analysis showed that virological failure was related to marital status and cotrimoxazole use before antiviral treatment. Among them, marriage and cohabitation [AHR: 0.69 (0.53, 0.90), P = 0.007] and no use of cotrimoxazole before antiviral treatment [AHR: 0.64 (0.47, 0.86), P = 0.003] were protective factors for virological failure (S3 Table). Using local polynomial regression (LOESS) to fit the relationship between different baseline BMI categories and viral load (Fig 4), it can be seen that the viral load of patients who were underweight in the later period (≥6 months) still exceeded 400 copies/ml. Cox regression was used to evaluate the association between baseline BMI and poor viral suppression (S4 Table and S1 Fig). Taking the normal BMI group as the reference group, underweight in HIV patients was a risk factor for poor viral suppression [AHR: 1.24 (1.04, 1.48), P = 0.016].

### Discussion

In our study, we analysed data from 6139 patients who started or received antiviral therapy in Guangxi with a change in the mean level of CD4+ T cells and a change in CD4+ T-cell gain

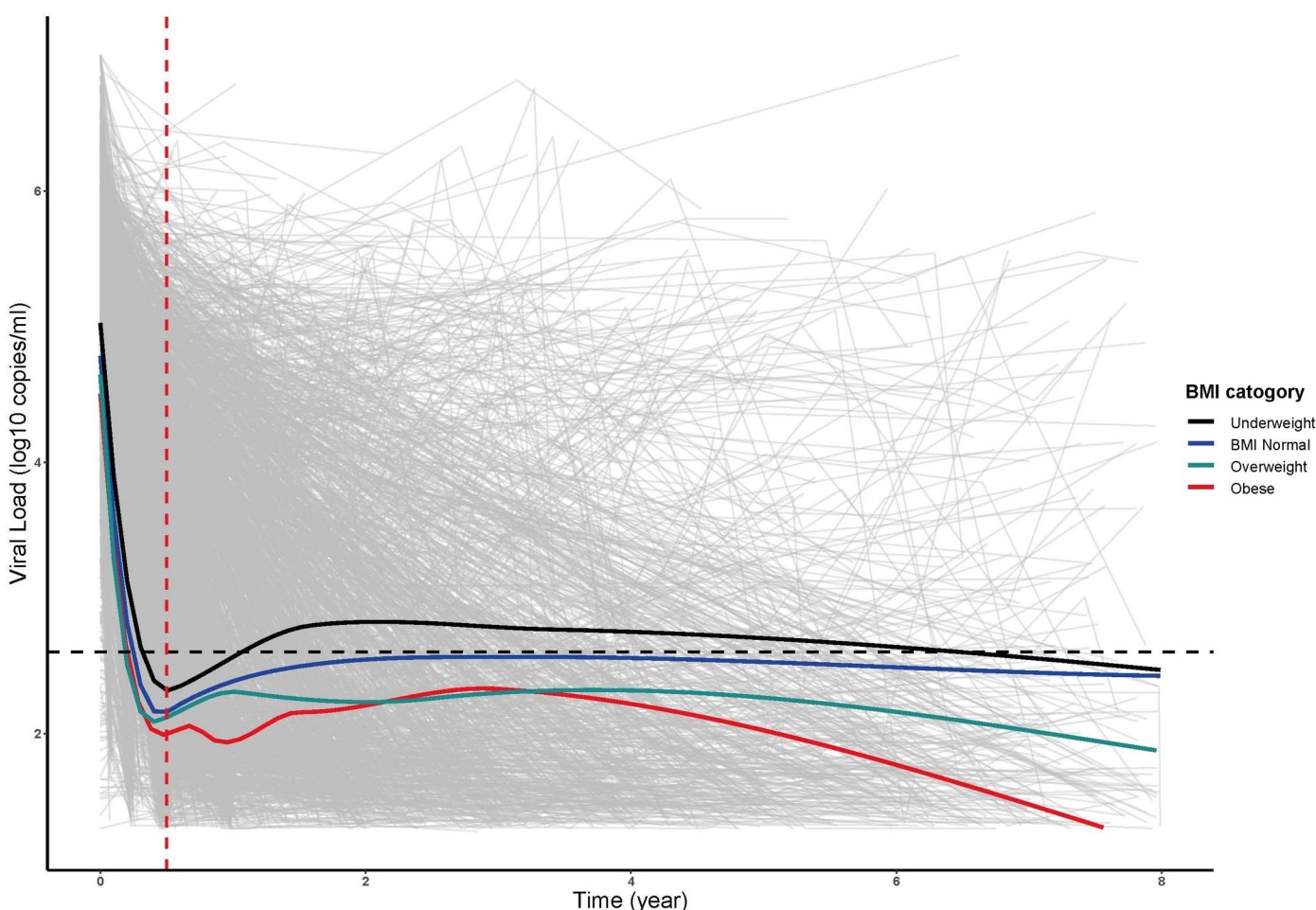

**Fig 4. The fitting curve of viral load changes in different BMI categories (the black dotted line indicates that the viral load is 400 copies/ml; red dotted line indicates the 6th month of follow-up).**

from the initial to the eighth year of treatment. We found that the higher the baseline BMI was, the better the effect of immune reconstitution was for patients undergoing ART. The gain in CD4+ T cells increased with baseline BMI, and with a longer follow-up, the gain in CD4+ T cells in obese patients changed significantly [24, 25].

CD4+ T cells continued to increase with the length of follow-up, especially in the first year, when the rate of CD4+ T-cell growth was most significant, a finding that is consistent with other studies [26, 27]. The gain level of CD4+ T cells in overweight patients was not significantly different from that in normal BMI patients in the first year, while the gain level of CD4 + T cells in obese patients was higher than that in normal BMI patients in our study. The increase may be due to different underlying mechanisms behind early CD4 increases, and the overweight and obese patients in the third and fifth years later had more significant CD4 + gain than the normal BMI group. However, a study report from the U.S. The Military HIV Natural History Study (NHS) [28] showed that obese patients had a smaller gain of CD4+ T cells than normal weight patients, suggesting that excess body weight may adversely affect immune recovery. Similarly, an observational cohort study at a clinic affiliated with Vanderbilt University in Nashville, USA [15] concluded that there is an optimal BMI range for immune reconstitution in patients receiving ART, with patients with a baseline BMI of 25–30 kg/m$^2$ achieving the greatest CD4+ T-cell gain at month 12, above which CD4+ T-cell gain decreases. In contrast, a cohort analysis of PLHIV from the Asia-Pacific region [24] found that compared with normal BMI (18.5–23 kg/m$^2$), patients with a BMI >27.5 kg/m$^2$ at baseline had a 28.8 cell/μL (95% CI: 6.6, 50.9 cell/μL) increase in CD4+ cell count. In addition, we also explored the association between BMI and immune recovery to the threshold in patients with baseline CD4+ T-cell counts <200 cells/μL and concluded that higher baseline BMI supported immune recovery to the threshold (CD4+ T cells ≥ 350 cells/μL, ≥500 cells/μL) during follow-up. Kaplan–Meier plots showed that the time required for immune recovery was shorter in obese patients (the median time of CD4+ T cell count of ≥350 cells/μL recovery was 4.42 years; the median time of CD4+ T cell count of ≥500 cells/μL recovery was 5.50 years). Using a trend test, the risk of immune reconstitution returning to the threshold was greater with increasing BMI category (p for trend < 0.05). The results of this study are consistent with the results of cohort analysis in the outpatient clinic of the First Affiliated Hospital of China Medical University in Shenyang [29].

Although immune reconstitution is a major goal of ART, it is unclear why higher baseline BMI levels positively affect CD4+ T-cell gain, and our study shows that peripheral CD4+ T cells in HIV/AIDS patients receiving ART are associated with the potential relevance of fat metabolism, which is critical for a better understanding of the biological determinants of immune reconstitution, especially in patients with poor immune reconstitution and underweight. Current studies have speculated that the link between BMI and immune reconstitution may be related to adipokines such as leptin, differences in thymus size, differences in lymphocyte populations in the gastrointestinal tract and other mucosal sites differences in T lymphocyte apoptosis [30, 31]. Some studies have pointed out that leptin may be related to the immune reconstitution mechanism of HIV-infected patients with higher BMI; that is, serum leptin concentration is positively correlated with body fat percentage [32]. Leptin is an inflammatory factor with multiple roles in the immune system, activating both adaptive and innate immunity [19, 33]. Overweight or obese individuals have higher serum leptin levels than normal-weight individuals [34]. Obesity adversely affects immune function in the general population [35]. However, for HIV-infected patients, obesity may have a protective effect on immune reconstitution. It may be that higher leptin levels promote immune recovery. Studies have shown that leptin can promote thymopoiesis, regulating T-cell immune responses. Thymocytes and peripheral CD4+ T cells can be induced in mice [36]. Studies have found that

adipocytes may be involved in the persistence of HIV [37], but no association of baseline BMI with virological failure was found in this study. Interestingly, underweight HIV patients had suboptimal virological suppression, consistent with a study of viral suppression in adults treated with antiviral therapy in Ethiopia [38], and malnutrition was an independent risk factor for unsuppressed viral infection load.

Our study benefits from a large sample, but there are still some limitations. First, because it was a retrospective cohort study, there may have been selection bias, resulting in significant differences in baseline characteristics. Second, there were fewer obese people in the study, approximately 2%. However, a sensitivity analysis showed the positive effect of BMI and immune reconstitution in obese patients. Third, this study only illustrates the association between BMI level and immune reconstitution but cannot prove a causal relationship between the two. Finally, these data lack follow-up BMI values, and thus, no further analysis of the relationship between updated BMI and immune reconstitution is provided.

## Supporting information

**S1 Checklist. STROBE statement—checklist of items that should be included in reports of observational studies.**
(DOCX)

**S1 Fig. Association of baseline BMI with poor viral suppression.**
(PDF)

**S1 Table. Univariate and multivariate analysis of CD4 change of HIV positive and AIDS with or without symptoms.**
(DOCX)

**S2 Table. Association between BMI and CD4 lymphocyte count recovery($\geq$500cells/μL) in different models.**
(DOCX)

**S3 Table. Factors associated with virologic failure.**
(DOCX)

**S4 Table. Factors associated with poor viral suppression.**
(DOCX)

## Acknowledgments

We sincerely thank all the study participants.

## Author Contributions

**Conceptualization:** Jiawen Zhu, Shanfang Qin, Chuanyi Ning.

**Data curation:** Haimei Huang.

**Formal analysis:** Jiawen Zhu.

**Funding acquisition:** Chuanyi Ning.

**Investigation:** Haimei Huang, Min Wang, Yun Zhang, Jinli Mo, Weiyi Tian, Sumin Tan, Li Jiang, Chuanyi Ning.

**Methodology:** Jiawen Zhu, Min Wang, Yun Zhang, Jinli Mo, Weiyi Tian, Sumin Tan, Li Jiang.

**Project administration:** Chuanyi Ning.

**Resources:** Chuanyi Ning.

**Software:** Jiawen Zhu.

**Supervision:** Min Wang, Yun Zhang, Jinli Mo, Weiyi Tian, Sumin Tan, Li Jiang, Zhihao Meng, Shanfang Qin, Chuanyi Ning.

**Validation:** Haimei Huang, Min Wang, Yun Zhang, Jinli Mo, Weiyi Tian, Sumin Tan, Li Jiang, Zhihao Meng, Shanfang Qin.

**Visualization:** Jiawen Zhu.

**Writing – original draft:** Jiawen Zhu, Haimei Huang.

**Writing – review & editing:** Zhihao Meng, Shanfang Qin, Chuanyi Ning.

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
