## [Decision Letter · Decision Letter 0]

12 Sep 2022

PONE-D-22-20667High baseline body mass index predicts recovery of CD4+ T lymphocytes for HIV/AIDS patients receiving long-term antiviral therapyPLOS ONE

Dear Dr. Ning,

Thank you for submitting your manuscript to PLOS ONE. After careful consideration, we feel that it has merit but does not fully meet PLOS ONE’s publication criteria as it currently stands. Therefore, we invite you to submit a revised version of the manuscript that addresses the points raised during the review process.

 The reviewers have made some comments and suggestions to improve the manuscript. Kindly consider all comments and revert with the revised manuscript timely.

We look forward to receiving your revised manuscript.

Kind regards,

Chika Kingsley Onwuamah, Ph.D.

Academic Editor

PLOS ONE

Journal Requirements:

"This study received support from the National Natural Science Foundation of China (Grant Nos. 81803295 and 81760602), the Natural Science Foundation of Guangxi (2018GXNSFAA138031), the “Thirteenth Five-Year” National Major Science and Technology Projects (2018ZX10715008–002 and 2018ZX10302104–001), the Innovation Project of Guangxi Graduate Education (YCSW2021143), and the Opening topic fund of Guangxi Key Laboratory of AIDS Prevention and Treatment (No.gklapt 201902)."

"This study received support from the National Natural Science Foundation of China (Grant Nos. 81803295 and 81760602), the Natural Science Foundation of Guangxi (2018GXNSFAA138031), the “Thirteenth Five-Year” National Major Science and Technology Projects (2018ZX10715008–002 and 2018ZX10302104–001), the Innovation Project of Guangxi Graduate Education (YCSW2021143), and the Opening topic fund of Guangxi Key Laboratory of AIDS Prevention and Treatment (No.gklapt 201902)."

Reviewers' comments:

Reviewer's Responses to Questions

**Comments to the Author**

1. Is the manuscript technically sound, and do the data support the conclusions?

Reviewer #1: Yes

Reviewer #2: Yes

2. Has the statistical analysis been performed appropriately and rigorously? 

Reviewer #1: Yes

Reviewer #2: Yes

3. Have the authors made all data underlying the findings in their manuscript fully available?

Reviewer #1: No

Reviewer #2: Yes

4. Is the manuscript presented in an intelligible fashion and written in standard English?

Reviewer #1: Yes

Reviewer #2: Yes

5. Review Comments to the Author

Reviewer #1: Have the authors made all data underlying the findings in their manuscript fully available?

The results state that Most patient were followed for ≤8 years. Furthermore, the authors also present that CD4+ T cells continued to increase with the length of follow-up. The authors however did not present results from the follow up after 8 years in the univariate and multivariate regression models even though this was used for the Kaplan Meier curves. it would be important to review which variables remained statistically significant across 1, 3, 5 and also 8 years of follow up.

Reviewer #2: REVIEW FOR MANUSCRIPT TITLED “HIGH BASELINE BODY MASS INDEX PREDICTS RECOVERY OF CD4+ T LYMPHOCYTES FOR HIV/AIDS PATIENTS RECEIVING LONG-TERM ANTIVIRAL THERAPY”

In this manuscript, the authors performed a database review of HIV-positive individuals to ascertain the relationship between baseline BMI and CD4 T cell count recovery in patients who are receiving ARVs for a period of 17 years.

The long duration of the review provided a robust database to obtain data. This is a favorable aspect of the study. The statistical input was quite detailed. The findings from the work tend to suggest a direct relationship between CD4 T cell count and BMI ie patients with higher BMI tend to have better recovery of CD4+ lymphocytes Findings from this work will influence key aspects of clinical decision-making in the care and management of patients living with HIV

However, the authors should provide clarity on the following aspects:

1. The author should distinguish between patients who have developed AIDs based on Clinical and laboratory criteria and those who are HIV positive.

2. There should be a statistical sub-analysis based upon this classification. This will help to put findings from the study in a proper perspective. For instance, it should be clear from the analysis if the recovery of CD4 T cells and T lymphocytes is better in patients WITHOUT AIDs defining symptoms versus patients WITH AIDs defining symptoms.

6. PLOS authors have the option to publish the peer review history of their article (what does this mean?). If published, this will include your full peer review and any attached files.

Reviewer #1: **Yes: **Obiageli Onwusaka

Reviewer #2: **Yes: **OHIHOIN AIGBE GREGORY

---

## [Author Response · Author response to Decision Letter 0]

25 Oct 2022

Article Title: High baseline body mass index predicts recovery of CD4+ T lymphocytes for HIV/AIDS patients receiving long-term antiviral therapy

Dear editors and reviewers,

Thank you for your further feedback. Please see below for our responses to your comments and concerns, listed in their original order of appearance; your comments are shown in bold, while our responses are shown in regular font type.

As per before, if you have any further questions, please do not hesitate to let us know.

Sincerely, 

Chuanyi Ning

Editor’s comments:

Response:

Thank you for the comment. We have double checked and modified the format of the manuscript as required.

"This study received support from the National Natural Science Foundation of China (Grant Nos. 81803295 and 81760602), the Natural Science Foundation of Guangxi (2018GXNSFAA138031), the “Thirteenth Five-Year” National Major Science and Technology Projects (2018ZX10715008–002 and 2018ZX10302104–001), the Innovation Project of Guangxi Graduate Education (YCSW2021143), and the Opening topic fund of Guangxi Key Laboratory of AIDS Prevention and Treatment (No.gklapt 201902)."

Response:

Thank you for the critique. We have included the funding statement in the cover letter, as well as the updated manuscript.

"This study received support from the National Natural Science Foundation of China (Grant Nos. 81803295 and 81760602), the Natural Science Foundation of Guangxi (2018GXNSFAA138031), the “Thirteenth Five-Year” National Major Science and Technology Projects (2018ZX10715008–002 and 2018ZX10302104–001), the Innovation Project of Guangxi Graduate Education (YCSW2021143), and the Opening topic fund of Guangxi Key Laboratory of AIDS Prevention and Treatment (No.gklapt 201902)."

Response:

Thank you for the critique. We have included the Role of Funder statement in the cover letter, as well as the updated manuscript.

Response:

Thank you for the critique. We have moved the ethical statement to the Methods section.

Response:

Thank you for the critique. We have replaced the incorrect reference in the revised version. Please let us know if you still have any question. 

The details are as follows:

Original citation: [7]STD and AIDS Prevention and Control Center CCfDCaP. National Free HIV Antiretroviral Drug Treatment Handbook (4th Edition): People's Medical Publishing House; 2016.

Replaced Citation: [7][Chinese guidelines for diagnosis and treatment of HIV/AIDS (2018)]. Zhonghua nei ke za zhi. 2018;57(12):867-84.

Reviewers’ comments:

Reviewer #1: Have the authors made all data underlying the findings in their manuscript fully available?

Response:

Thank you for the critique. Many thanks to the editor for your efforts to this manuscript. We have the right to access the raw data and no administrative permissions is required. But the raw data is not publicly available because of ethical and legal reasons. 

The results state that Most patient were followed for ≤8 years. Furthermore, the authors also present that CD4+ T cells continued to increase with the length of follow-up. The authors however did not present results from the follow up after 8 years in the univariate and multivariate regression models even though this was used for the Kaplan Meier curves. it would be important to review which variables remained statistically significant across 1, 3, 5 and also 8 years of follow up.

Response:

Thank you for the critique. Most patients were followed up for ≤8 years, 114 patients (about 1.9%) were followed up for more than 8 years, and only 79 patients were able to calculate the change of CD4 lymphocyte count. In order to avoid the low efficiency of statistical test, our analysis data were not included in the patients who had been followed up for more than 8 years. Therefore, we supplemented the 8-year follow-up data to the updated table 2 according your suggestions. The data showed that age, the occurrence of virologic failure, and sex were related to the change in CD4+ T cells during the 1st, 3rd, 5th and 8th years of follow-up. Taking the fifth year as an example, older age [-3.36 (-4.06, -2.65), P<0.001] and the occurrence of virologic failure [-115.81 (-150.32, -81.29), P<0.001] were negative factors for CD4+ T-cell gain. CD4+ T-cell concentration was significantly increased in females compared with males [48.28 (30.15, 66.41), P<0.001].

Reviewer #2: REVIEW FOR MANUSCRIPT TITLED “HIGH BASELINE BODY MASS INDEX PREDICTS RECOVERY OF CD4+ T LYMPHOCYTES FOR HIV/AIDS PATIENTS RECEIVING LONG-TERM ANTIVIRAL THERAPY”

In this manuscript, the authors performed a database review of HIV-positive individuals to ascertain the relationship between baseline BMI and CD4 T cell count recovery in patients who are receiving ARVs for a period of 17 years.

The long duration of the review provided a robust database to obtain data. This is a favorable aspect of the study. The statistical input was quite detailed. The findings from the work tend to suggest a direct relationship between CD4 T cell count and BMI ie patients with higher BMI tend to have better recovery of CD4+ lymphocytes Findings from this work will influence key aspects of clinical decision-making in the care and management of patients living with HIV

However, the authors should provide clarity on the following aspects:

1. The author should distinguish between patients who have developed AIDs based on Clinical and laboratory criteria and those who are HIV positive.

Response:

Thank you very much for pointing this out. We have added the analysis to compare the CD4 changes between the HIV positive, AIDS with AIDS defining symptoms and AIDS without AIDS defining symptoms according your comment #1 and comment #2. The data analysis results showed that there was no correlation between each group for CD4 changes, and the difference was not statistically significant. We added these results in the supplementary materials, please see the Table S1. Please let us know if you still have question.

Table S1. Univariate and Multivariate Analysis of CD4 Change of HIV Positive and AIDS with or without Symptoms 

2.There should be a statistical sub-analysis based upon this classification. This will help to put findings from the study in a proper perspective. For instance, it should be clear from the analysis if the recovery of CD4 T cells and T lymphocytes is better in patients WITHOUT AIDs defining symptoms versus patients WITH AIDs defining symptoms.

Response:

Thank you very much for your comments. As mentioned above, please see the Table S1 above. We selected patients defined as AIDS, and divided them into two groups according to whether they have AIDS defining symptoms or not. We used a linear regression model to analyze and compare the changes of CD4 in patients without AIDS defining symptoms versus patients with AIDS defining symptoms. Results also shown that there was no statistical difference between these two groups. Please see more details in the Table S1.

---

## [Decision Letter · Decision Letter 1]

8 Dec 2022

PONE-D-22-20667R1High baseline body mass index predicts recovery of CD4+ T lymphocytes for HIV/AIDS patients receiving long-term antiviral therapyPLOS ONE

Dear Dr. Ning,

Thank you for submitting your manuscript to PLOS ONE. After careful consideration, we feel that it has merit but does not fully meet PLOS ONE’s publication criteria as it currently stands. Therefore, we invite you to submit a revised version of the manuscript that addresses the points raised during the review process.

Please see below the few issues raised by the reviewers and kindly address them point-by-point.

We look forward to receiving your revised manuscript.

Kind regards,

Chika Kingsley Onwuamah, Ph.D.

Academic Editor

PLOS ONE

Journal Requirements:

Reviewers' comments:

Reviewer's Responses to Questions

**Comments to the Author**

1. If the authors have adequately addressed your comments raised in a previous round of review and you feel that this manuscript is now acceptable for publication, you may indicate that here to bypass the “Comments to the Author” section, enter your conflict of interest statement in the “Confidential to Editor” section, and submit your "Accept" recommendation.

Reviewer #1: All comments have been addressed

Reviewer #2: All comments have been addressed

2. Is the manuscript technically sound, and do the data support the conclusions?

Reviewer #1: Yes

Reviewer #2: Yes

3. Has the statistical analysis been performed appropriately and rigorously? 

Reviewer #1: Yes

Reviewer #2: Yes

4. Have the authors made all data underlying the findings in their manuscript fully available?

Reviewer #1: Yes

Reviewer #2: Yes

5. Is the manuscript presented in an intelligible fashion and written in standard English?

Reviewer #1: Yes

Reviewer #2: Yes

6. Review Comments to the Author

Reviewer #1: (No Response)

Reviewer #2: REVIEW FOR MANUSCRIPT TITLED “HIGH BASELINE BODY MASS INDEX PREDICTS RECOVERY OF CD4+ T LYMPHOCYTES FOR HIV/AIDS PATIENTS RECEIVING LONG-TERM ANTIVIRAL THERAPY”

In this manuscript, the authors performed a database review of HIV-positive individuals to ascertain the relationship between baseline BMI and CD4 T cell count recovery in patients who are receiving ARVs for a period of 17 years.

The long duration of the review provided a robust database to obtain data. This is a favorable aspect of the study. The statistical input was quite detailed. The findings from the work tend to suggest a direct relationship between CD4 T cell count and BMI ie patients with higher BMI tend to have better recovery of CD4+ lymphocytes Findings from this work will influence key aspects of clinical decision-making in the care and management of patients living with HIV

However, the authors should provide clarity on the following aspects:

1. The author should distinguish between patients who have developed AIDs based on Clinical and laboratory criteria and those who are HIV positive.

2. There should be a statistical sub-analysis based upon this classification. This will help to put findings from the study in a proper perspective. For instance, it should be clear from the analysis if the recovery of CD4 T cells and T lymphocytes is better in patients WITHOUT AIDs defining symptoms versus patients WITH AIDs defining symptoms.

UPDATE ON REVIEW

The authors have responded appropriately to my earlier comments above.

7. PLOS authors have the option to publish the peer review history of their article (what does this mean?). If published, this will include your full peer review and any attached files.

Reviewer #1: **Yes: **Obiageli Chiezey Onwusaka

Reviewer #2: **Yes: **OHIHOIN AIGBE GREGORY

---

## [Author Response · Author response to Decision Letter 1]

12 Dec 2022

Article Title: High baseline body mass index predicts recovery of CD4+ T lymphocytes for HIV/AIDS patients receiving long-term antiviral therapy

Dear editors and reviewers,

Thank you for your further feedback. Please see below for our responses to your comments and concerns, listed in their original order of appearance; your comments are shown in bold, while our responses are shown in regular font type.

As per before, if you have any further questions, please do not hesitate to let us know.

Sincerely, 

Chuanyi Ning

Editor’s comments:

Journal Requirements:

Response:

Thank you for the critique. We have updated the reference style in the revised edition and replaced the incorrect reference. Please let us know if you still have any question. 

The details are as follows:

Original citation: [34]Sørensen TI, Echwald S, Holm JC. Leptin in obesity. BMJ (Clinical research ed). 1996;313(7063):953-4.

Replaced Citation: [34]Zimmet P, Hodge A, Nicolson M, Staten M, de Courten M, Moore J, et al. Serum leptin concentration, obesity, and insulin resistance in Western Samoans: cross sectional study. BMJ (Clinical research ed). 1996;313(7063):965-9. Epub 1996/10/19. doi: 10.1136/bmj.313.7063.965. PubMed PMID: 8892415; PubMed Central PMCID: PMCPMC2352310.

We have fully dealt with the comments of the two reviewers in the last round of review and have been recognized by the two reviewers. Here we have updated the references and hope to get your reply.

---

## [Editor Report · Decision Letter 2]

14 Dec 2022

High baseline body mass index predicts recovery of CD4+ T lymphocytes for HIV/AIDS patients receiving long-term antiviral therapy

PONE-D-22-20667R2

Dear Dr. Ning,

We’re pleased to inform you that your manuscript has been judged scientifically suitable for publication and will be formally accepted for publication once it meets all outstanding technical requirements.

Kind regards,

Chika Kingsley Onwuamah, Ph.D.

Academic Editor

PLOS ONE
---

## [Editor Report · Acceptance letter]

19 Dec 2022

PONE-D-22-20667R2 

High baseline body mass index predicts recovery of CD4+ T lymphocytes for HIV/AIDS patients receiving long-term antiviral therapy 

Dear Dr. Ning:

I'm pleased to inform you that your manuscript has been deemed suitable for publication in PLOS ONE. Congratulations! Your manuscript is now with our production department. 

Kind regards, 

on behalf of

Dr. Chika Kingsley Onwuamah 

Academic Editor

PLOS ONE